# Neural Interpretable PDEs: Harmonizing Fourier Insights with Attention for Scalable and Interpretable Physics Discovery

**Ning Liu** [1 2]  **Yue Yu** [1]

## Abstract

Attention mechanisms have emerged as transformative tools in core AI domains such as natural language processing and computer vision. Yet, their largely untapped potential for modeling intricate physical systems presents a compelling frontier. Learning such systems often entails discovering operators that map between functional spaces using limited instances of function pairs—a task commonly framed as a severely ill-posed inverse PDE problem. In this work, we introduce Neural Interpretable PDEs (NIPS), a novel neural operator architecture that builds upon and enhances Nonlocal Attention Operators (NAO) in both predictive accuracy and computational efficiency. NIPS employs a linear attention mechanism to enable scalable learning and integrates a learnable kernel network that acts as a channel-independent convolution in Fourier space. As a consequence, NIPS eliminates the need to explicitly compute and store large pairwise interactions, effectively amortizing the cost of handling spatial interactions into the Fourier transform. Empirical evaluations demonstrate that NIPS consistently surpasses NAO and other baselines across diverse benchmarks, heralding a substantial leap in scalable, interpretable, and efficient physics learning. Our code and data accompanying this paper are available at https://github.com/fishmoon1234/Nonlocal-Attention-Operator.

## 1. Introduction

Interpretability in machine learning (ML) models is of paramount importance, particularly in the study of physical systems, where grasping the underlying governing principles is as critical as achieving high predictive accuracy. Unlike conventional data-driven tasks, physics-based problems necessitate models that not only yield reliable outputs but also elucidate the fundamental mechanisms driving observed phenomena. This need is especially acute in high-stakes domains such as materials science, healthcare, and engineering, where ML-driven insights can directly influence human safety and technological progress (Coorey et al., 2022; Ferrari & Willcox, 2024; Liu et al., 2025). However, extracting physical laws from data remains a formidable challenge, as many learning tasks—such as deducing material properties from deformation fields—are inherently an ill-posed inverse problem, lacking direct supervision and admitting multiple plausible solutions (Hansen, 1998). That means, although ML models may serve as effective surrogates for predictive tasks, their inferred internal representations may diverge from true physical parameters, leading to spurious interpretations and potential failures in practical applications. Thus, advancing interpretability is not merely a matter of transparency but an essential endeavor in ensuring the development of robust, trustworthy, and scientifically principled ML models for physics-based learning.

Uncovering interpretable mechanisms for physical systems presents a fundamental challenge: inferring governing laws that are often high- or infinite-dimensional from data consisting of discrete measurements of continuous functions. A data-driven surrogate model must therefore learn not only the mapping between input-output function pairs but also the latent representations that govern the system's behavior. From a partial differential equation (PDE) perspective, constructing a surrogate model corresponds to solving a forward problem, whereas identifying the underlying governing mechanism constitutes an inverse problem, which is often rank-deficient or even ill-posed, especially when data is limited. Neural network models exacerbate this ill-posedness due to their intrinsic approximation biases (Xu et al., 2019), making the inference of governing laws particularly challenging. To address this, recent deep learning

[1]Department of Mathematics, Lehigh University, Bethlehem, PA 18015, USA [2]Global Engineering and Materials, Inc., Princeton, NJ 08540, USA. Correspondence to: Yue Yu <yuy214@lehigh.edu>.

*Proceedings of the 42nd International Conference on Machine Learning*, Vancouver, Canada. PMLR 267, 2025. Copyright 2025 by the author(s).

approaches have been developed as inverse problem solvers (Fan & Ying, 2023; Molinaro et al., 2023; Jiang et al., 2022; Chen et al., 2023; Lu & Yu, 2025), aiming to reconstruct unknown parameters from solution data. These approaches generally rely on effectively embedding prior information, in terms of governing equations (Yang et al., 2021; Li et al., 2021), regularization techniques (Dittmer et al., 2020; Obmann et al., 2020; Ding et al., 2022; Chen et al., 2023), or additional operator structures (Uhlmann, 2009; Lai et al., 2019; Yilmaz, 2001). However, in complex systems, such priors are often unavailable or highly problem-specific, limiting the generalizability of these approaches. Consequently, existing methods can only solve inverse problems for a fixed system, requiring a complete reconfiguration when the system evolves—such as when material properties degrade in a material modeling task.

In this work, we introduce **Neural Interpretable PDEs (NIPS)**, a novel attention-based neural operator architecture that extends and enhances Nonlocal Attention Operators (NAO) (Yu et al., 2024) in both predictive accuracy and computational efficiency. NAO introduces an attention-based kernel map to simultaneously learn the operator and its associated kernel, effectively extracting hidden knowledge from diverse physical systems. By implicitly enforcing prior information and exploring the function space of identifiability for the kernels, the attention mechanism enables automatic inference of underlying system contexts in an unsupervised manner (Lu & Yu, 2025). Nevertheless, NAO relies on an attention mechanism of quadratic complexity and explicitly models spatial interactions through a square projection matrix, which are both time- and memory-intensive. In contrast, NIPS adopts a linear attention mechanism for scalable learning and incorporates a learnable kernel network that functions as a channel-independent convolution in Fourier space. This design eliminates the need to compute and store large pairwise interactions explicitly, effectively amortizing the cost of spatial interactions through the Fourier transform. **Our key contributions** are:

- We introduce Neural Interpretable PDEs (NIPS), a novel attention-based neural operator architecture for simultaneous physics modeling (as a forward PDE solver) and governing physical mechanism discovery (as an inverse PDE solver).

- NIPS integrates attention with a learnable kernel network for channel-independent convolution in Fourier space, eliminating the need to compute and store large pairwise interactions and amortizing the cost of handling spatial interactions into the Fourier transform.

- We reformulate the original NAO using linear attention, enabling scalable learning in large physical systems. Together with Fourier convolution, NIPS effectively harmonizes Fourier insights with attention for scalable and interpretable physics discovery.

- We conduct zero-shot learning experiments on unseen physical systems, demonstrating both the generalizability of NIPS in predicting system responses and its capability in discovering hidden PDE parameters.

## 2. Background and Related Work

**Neural operators for hidden physics learning.** Learning complex physical systems from data has become increasingly prevalent in scientific and engineering fields (Karniadakis et al., 2021; Liu et al., 2024b; Ghaboussi et al., 1991; Carleo et al., 2019; Zhang et al., 2018; Liu et al., 2023b; Cai et al., 2022; Pfau et al., 2020). In many cases, the underlying governing laws remain unknown, concealed within the data, and must be uncovered through physical models. Ideally, these models should be *interpretable*, enabling domain experts to derive meaningful insights and make further predictions, thereby advancing the understanding of the target physical system (Jafarzadeh et al., 2024; Wang et al., 2025). Furthermore, they should be *resolution-invariant*, capable of handling data at different scales. Neural operators (NOs) are designed to map between infinite-dimensional function spaces (Li et al., 2020a;c; You et al., 2022a; Ong et al., 2022; Cao, 2021; Lu et al., 2019; 2021; Goswami et al., 2022; Liu et al., 2024a), making them a powerful tool for discovering continuum physical laws by capturing the relationships between spatial and spatio-temporal data. A major milestone in operator learning is the Fourier Neural Operator (FNO) (Li et al., 2020c; Liu et al., 2023a), which utilizes the Fast Fourier Transform (FFT) to efficiently perform convolutions in Fourier space. This approach reduces computational complexity, enabling scalable learning of high-dimensional physical systems while retaining the capacity to capture intricate physical interactions. By incorporating FFT, NOs can learn kernels with truncated modes in Fourier space, enhancing efficiency and making them a promising tool for complex modeling physical systems across various scales.

**Inverse PDE solving.** Inverse PDE solving involves uncovering the underlying physical laws or parameters from observational data (Liu et al., 2024c). This task is particularly challenging due to the high dimensionality of the problem and the need to infer unknowns from limited or noisy data. Recent progress in inverse PDE solving has led to the development of innovative approaches (Cho & Son, 2024), such as Neural Inverse Operators (NIOs) (Molinaro et al., 2023) and Nonlocal Attention Operators (NAOs) (Yu et al., 2024). NIOs are designed to map solution operators to the underlying PDE parameters as functions in a supervised setting, leveraging a composition of DeepONets and FNOs. This architecture has demonstrated superior performance in solving inverse PDE problems, outperforming traditional optimization methods. On the other hand, NAOs employ attention mechanisms to capture nonlocal interactions among spatial tokens in an unsupervised setting. They

effectively mitigate ill-posedness and rank deficiency by implicitly extracting global prior knowledge from training data of multiple systems (Lu & Yu, 2025). NAOs can generalize across unseen data resolutions and system states, demonstrating their potential as interpretable foundation models for physical systems.

**Attention mechanism.** Attention mechanisms (Vaswani et al., 2017) are renowned for their remarkable in-context learning capabilities in language models. In particular, they enable zero-shot generalizability, allowing models to adapt to new tasks from just a few examples in the prompt without retraining (Vladymyrov et al., 2024; Lu et al., 2024; Oko et al., 2024). Recently, attention mechanisms have become integral to learning hidden physics, especially in tasks involving PDE solving. Beyond generalizability, attention mechanisms enhance model capacity by capturing long-range dependencies and spatiotemporal interactions, making them highly effective for modeling complex physical systems. To improve the accuracy of forward PDE solvers, linear attention has been employed in NOs to replace softmax normalization and act as a learnable kernel (Cao, 2021). This relaxes the quadratic complexity requirement while maintaining the ability to capture intricate physical relationships. Further innovations include the integration of Galerkin-type linear attention within encoder-decoder architectures (Li et al., 2022), hierarchical transformers for multiscale learning (Liu et al., 2022), and heterogeneous normalized attention mechanisms for handling multiple input features (Hao et al., 2023). On a related note, polynomial attention (Kacham et al., 2023; Deng et al., 2024) offers a promising alternative to linear attention by replacing the softmax function with polynomial kernels, enabling efficient computation of attention scores, particularly for long-range dependencies and complex physical systems. Recent studies have applied attention mechanisms not only to solve individual PDEs but also to tackle multiple types of PDEs within a unified framework, demonstrating the potential of these methods in advancing interpretable and scalable models for physical mechanism discovery (Yang & Osher, 2024; Ye et al., 2024; Sun et al., 2024; Zhang, 2024).

# 3. Neural Interpretable PDEs

This section introduces NIPS, a Fourier-domain kernel operator that simultaneously solves both forward and inverse PDE problems. Compared to existing NOs, NIPS incorporates two key innovations. First, it constructs a data-dependent neural operator through the linear attention mechanism, parameterizing an inverse mapping from data to the underlying kernel and offering a generalizable kernel method for PDEs with varying hidden parameter fields. Second, the attention mechanism is combined with spectral convolution to enhance expressivity by enabling interactions across different frequency modes, while also providing efficient evaluation of domain integrals. As a result, NIPS delivers an accurate, efficient, and interpretable PDE solution operator.

## 3.1. Problem Setting

We consider a series of PDEs with different hidden physical parameters:

$$\mathcal{L}_{\boldsymbol{b}}[\boldsymbol{u}](\boldsymbol{x}) = \boldsymbol{f}(\boldsymbol{x}) , \quad \boldsymbol{x} \in \Omega . \tag{1}$$

Here, $\Omega \subset \mathbb{R}^s$ is the domain of interest, $\boldsymbol{f}(\boldsymbol{x})$ represents the loading function on $\Omega$, and $\boldsymbol{u}(\boldsymbol{x})$ is the corresponding solution of the system. $\mathcal{L}_{\boldsymbol{b}}$ denotes the unknown governing law, such as balance laws, which is determined by the (possibly unknown and high-dimensional) parameter field $\boldsymbol{b}$. For example, in material modeling, $\mathcal{L}_{\boldsymbol{b}}$ typically represents the constitutive law, and $\boldsymbol{b}$ can be a vector ($\boldsymbol{b} \in \mathbb{R}^{d_b}$) representing the homogenized material parameter field, or a vector-valued function ($\boldsymbol{b} \in L^{\infty}(\Omega; \mathbb{R}^{d_b})$) representing the heterogeneous material properties.

Many physical modeling tasks can be formulated as either forward or inverse PDE-solving problems. In a forward problem setting, the goal is to find the PDE solution given the PDE information, such as coefficient functions, boundary/initial conditions, and loading sources. Concretely, given the governing operators $\mathcal{K}$, the parameter (field) $\boldsymbol{b}$, and loading field $\boldsymbol{f}(\boldsymbol{x})$ in (1), the objective is to solve for the corresponding solution field $\boldsymbol{u}(\boldsymbol{x})$ using either classical PDE solvers (Brenner & Scott, 2007) or data-driven approaches (Lu et al., 2019; Li et al., 2020c). As a result, a forward map is constructed:

$$\mathcal{G} : (\boldsymbol{b}, \boldsymbol{f}) \to \boldsymbol{u} . \tag{2}$$

Here, $\boldsymbol{b}$ and $\boldsymbol{f}$ are input vectors/functions, and $\boldsymbol{u}$ is the output function.

Conversely, solving an inverse PDE problem involves reconstructing the underlying full or partial PDE information from solutions, where one seeks to construct an inverse map:

$$\mathcal{H} : (\boldsymbol{u}, \boldsymbol{f}) \to \boldsymbol{b} . \tag{3}$$

Solving an inverse problem is typically more challenging due to the ill-posed nature of the PDE model. In general, a limited number of function pairs $(\boldsymbol{u}, \boldsymbol{f})$ from a single system is insufficient for inferring the underlying parameter field $\boldsymbol{b}$, rendering the inverse problem generally non-identifiable (Molinaro et al., 2023).

Formally, we consider $S$ training datasets from different systems, each of which contains $d$ function pairs $(\boldsymbol{u}, \boldsymbol{f}) \in \mathcal{U} \times \mathcal{F}$:

$$\mathcal{D}_{\mathrm{tr}} = \{\{(\boldsymbol{u}_i^{\eta}(\boldsymbol{x}), \boldsymbol{f}_i^{\eta}(\boldsymbol{x}))\}_{i=1}^{d}\}_{\eta=1}^{S} . \tag{4}$$

The index $\eta$ corresponds to different hidden PDE parameter field/set $\boldsymbol{b}^{\eta} \in \mathcal{B}$. Here, $\mathcal{U} \subset L^2(\Omega; \mathbb{R}^{d_u})$, $\mathcal{F} \subset L^2(\Omega; \mathbb{R}^{d_f})$ represent the (infinite-dimensional) Banach spaces of the

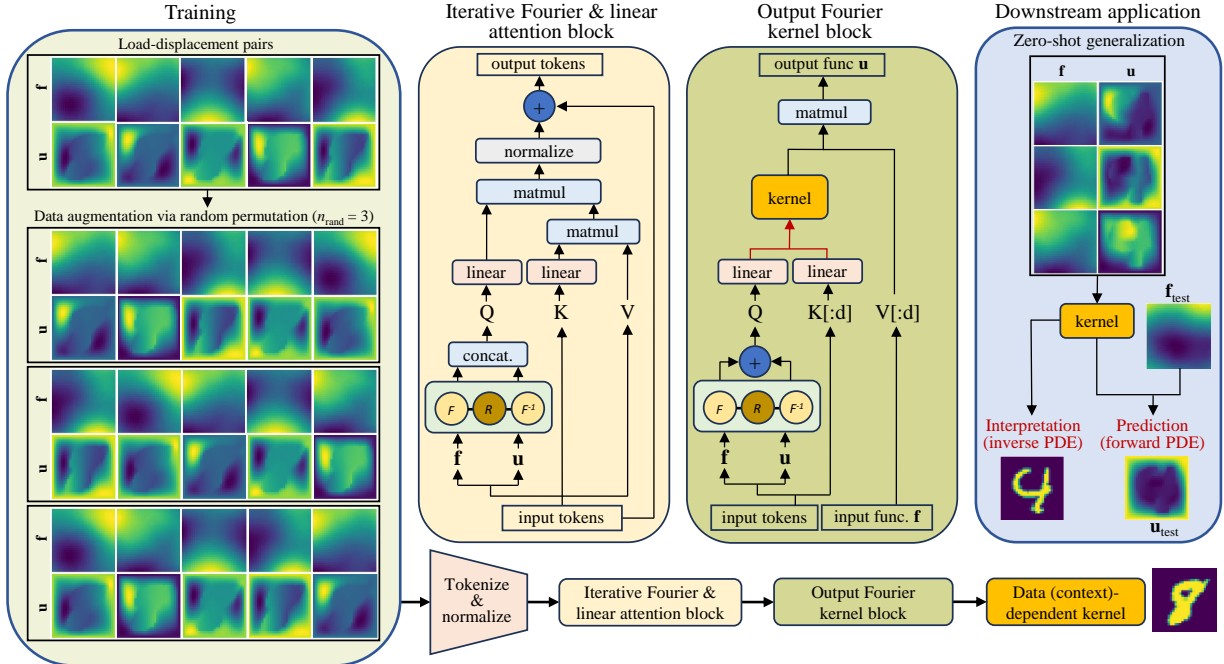

*Figure 1.* Illustration of the NIPS architecture. NIPS applies physics-informed data augmentation by randomly permuting embedding dimensions to prevent features from being tied to a specific sequence. The data is then tokenized and normalized, followed by several Fourier and linear attention layers that implicitly extract hidden prior knowledge from multiple physical systems. For each system, the final layer maps the last iterative layer feature from contextual input-output function pairs to the corresponding kernel. In downstream tasks, discovering the hidden kernel only requires a forward pass from contextual data, without requiring model retraining.

solution function $\boldsymbol{u}$ and loading function $\boldsymbol{f}$, respectively, and $\mathcal{B}$ is a finite-dimensional manifold where the hidden parameter $\boldsymbol{b}$ takes values on. Our goal is to solve both the forward and inverse PDE problems simultaneously from multiple systems. That means, given $\mathcal{D}_{\mathrm{tr}}$ and measurements $\{(\boldsymbol{u}_i^{test}(\boldsymbol{x}), \boldsymbol{f}_i^{test}(\boldsymbol{x}))\}_{i=1}^d$ corresponding to a new and hidden parameter field $\boldsymbol{b}^{test}$, we aim to infer the key information about $\boldsymbol{b}^{test}$ and provide a surrogate mapping $\mathcal{G}^{test} : \boldsymbol{f}_i^{test} \to \boldsymbol{u}_i^{test}$ without additional training or tuning.

### 3.2. Attention-Based Kernel Map

To develop a generalizable forward PDE solver capable of handling multiple PDEs, our architecture is inspired by the kernel method for PDEs (Evans, 2002). In this method, the solution of a linear PDE is formulated as an integral involving a kernel function that captures the influence of one point in space on another:

$$\mathcal{G}_{K_{\boldsymbol{b}}}[\boldsymbol{f}](\boldsymbol{x}) = \int_\Omega K_{\boldsymbol{b}}(\boldsymbol{x}, \boldsymbol{y})\boldsymbol{f}(\boldsymbol{y})d\boldsymbol{y} = \boldsymbol{u}(\boldsymbol{x}), \ \boldsymbol{x} \in \Omega. \quad (5)$$

Here, the kernel function depends on the parameter field $\boldsymbol{b}$. In generic integral NOs (Li et al., 2020b;c; You et al., 2022a;b) such as FNOs and GNOs, the kernel $K_{\boldsymbol{b}}$ is parameterized as either a convolutional kernel or a shallow MLP and is then optimized to minimize the data loss $\sum_{i=1}^d ||\mathcal{G}[\boldsymbol{f}_i; \theta] - \boldsymbol{u}_i||$ on function pairs from a single PDE.

Consequently, the forward solver $\mathcal{G}$ is tailored specifically for that particular PDE.

The first key feature of NIPS is the transition from a static kernel to a data-dependent kernel. In particular, we propose to replace $K_{\boldsymbol{b}}$ with a kernel map $K[\boldsymbol{u}_{1:d}, \boldsymbol{f}_{1:d}]$ in the form:

$$\mathcal{G}_{K_{\boldsymbol{b}}}[\boldsymbol{f}](\boldsymbol{x}) = \int_\Omega K[\boldsymbol{u}_{1:d}, \boldsymbol{f}_{1:d}](\boldsymbol{x}, \boldsymbol{y})\boldsymbol{f}(\boldsymbol{y})d\boldsymbol{y}, \quad (6)$$

where $\boldsymbol{x} \in \Omega$, and the nonlinear kernel map is constructed using $L$ numbers of iterative attention blocks:

$$\boldsymbol{g}_j^{(0)}(\boldsymbol{x}) := \boldsymbol{f}_j(\boldsymbol{x}), \ \boldsymbol{v}_j^{(0)}(\boldsymbol{x}) := \boldsymbol{u}_j(\boldsymbol{x}),$$

$$\begin{pmatrix} \boldsymbol{g}_j^{(l)}(\boldsymbol{x}) \\ \boldsymbol{v}_j^{(l)}(\boldsymbol{x}) \end{pmatrix} = \mathcal{N} \left( \int_\Omega K^{(l)}(\boldsymbol{x}, \boldsymbol{y}) \begin{pmatrix} \boldsymbol{g}_j^{(l-1)}(\boldsymbol{y}) \\ \boldsymbol{v}_j^{(l-1)}(\boldsymbol{y}) \end{pmatrix} d\boldsymbol{y} \right),$$

$$1 \le l \le L, \quad (7)$$

with

$$K^{(l)}(\boldsymbol{x}, \boldsymbol{y}) := \begin{bmatrix} K[\boldsymbol{g}^{(l)}, \boldsymbol{g}^{(l)}; \boldsymbol{W}_l] & K[\boldsymbol{g}^{(l)}, \boldsymbol{v}^{(l)}; \boldsymbol{W}_l] \\ K[\boldsymbol{v}^{(l)}, \boldsymbol{g}^{(l)}; \boldsymbol{W}_l] & K[\boldsymbol{v}^{(l)}, \boldsymbol{v}^{(l)}; \boldsymbol{W}_l] \end{bmatrix}, \quad (8)$$

$$K[\boldsymbol{p}, \boldsymbol{q}; \boldsymbol{W}] := \sum_{\omega, \nu=1}^d \int_\Omega \boldsymbol{W}^P(\boldsymbol{x}, \boldsymbol{z}) \left( \boldsymbol{p}_\omega(\boldsymbol{z}) \boldsymbol{W}^{QK}[\omega, \nu] \boldsymbol{q}_\nu(\boldsymbol{y}) \right) d\boldsymbol{z}.$$

Here, $\mathcal{N}$ is a chosen normalizer, $\boldsymbol{W}_l^{QK} := \frac{1}{\sqrt{d_k}} \boldsymbol{W}_l^Q(\boldsymbol{W}_l^K)^\top$ characterizes the (trainable) interaction of $d$ input function pair instances, $\boldsymbol{W}_l^P$ is a projection function, and $\theta := \{\{\boldsymbol{W}_l^P(\boldsymbol{x}, \boldsymbol{z}), \boldsymbol{W}_l^Q, \boldsymbol{W}_l^K\}_{l=1}^L\}$ are learnable

functions/parameters. By setting $\boldsymbol{v}_j^{(L)}(\boldsymbol{x}) := \boldsymbol{u}_j(\boldsymbol{x})$, we notice that the kernel $K[\boldsymbol{u}_{1:d}, \boldsymbol{f}_{1:d}]$ can be expressed as:

$$K[\mathbf{u}_{1:d}, \mathbf{f}_{1:d}; \theta](\boldsymbol{x}, \boldsymbol{y})$$

$$:= \sum_{\omega,\nu=1}^{d} \int_{\Omega} \boldsymbol{W}_L^{P,u}(\boldsymbol{x}, \boldsymbol{z}) \left( \boldsymbol{g}_\omega^{(L-1)}(\boldsymbol{z}) \boldsymbol{W}_L^{QK}[\omega,\nu] \boldsymbol{g}_\nu^{(L-1)}(\boldsymbol{y}) \right) d\boldsymbol{z}$$

$$+ \sum_{\omega,\nu=1}^{d} \int_{\Omega} \boldsymbol{W}_L^{P,f}(\boldsymbol{x}, \boldsymbol{z}) \left( \boldsymbol{v}_\omega^{(L-1)}(\boldsymbol{z}) \boldsymbol{W}_L^{QK}[\omega,\nu] \boldsymbol{g}_\nu^{(L-1)}(\boldsymbol{y}) \right) d\boldsymbol{z}.$$

Hence, the proposed architecture simultaneously serves as an inverse PDE solver, mapping $(\boldsymbol{u}, \boldsymbol{f})$ pairs to the parameter-dependent kernel $K_{\boldsymbol{b}}$, which provides a forward PDE solver in the form of a kernel method as in $\mathcal{G}_K$.

### 3.3. Efficient Fourier Kernel Learning

Although the attention block in (7) enables a generalizable kernel for multiple PDEs, it is inherently more computationally expensive than its single-PDE counterpart. Comparing (7) and (5), one can see that the single-PDE solver involves a spatial integral per layer update, whereas the proposed multi-PDE solver requires evaluating two integrals over the entire domain $\Omega$: the first integral in (8) maps data pairs to the kernel $K$, while the second performs the kernel-weighted integral in (7), updating the data pairs for the next block. When these integrals are computed on a discretized domain with $N$ grid points, both require $O(N^2 d)$ flops, making the model computationally expensive on large discretizations.

The class of shift-equivariant kernels can be decomposed as a linear combination of eigenfunctions, enabling efficient multiplication in the eigen-transform domain and accelerating integral evaluations. For single-PDE solvers, a notable example is the FNO model (Li et al., 2020c), which enforces translation invariance on the kernel $K_{\boldsymbol{b}}(\boldsymbol{x}, \boldsymbol{y}) := K_{\boldsymbol{b}}(\boldsymbol{x} - \boldsymbol{y})$ in (5), reducing the computational complexity from $O(N^2)$ to $O(N \log N)$. Inspired by this, the second core feature of NIPS enhances efficiency by enforcing a convolutional structure and computing the integral as element-wise multiplication in the frequency domain.

To alleviate the computational cost, we first notice that the proposed architecture can be reformulated from a linear attention perspective:

$$\int_\Omega K[\boldsymbol{p}, \boldsymbol{q}; \boldsymbol{W}](\boldsymbol{x}, \boldsymbol{y}) \boldsymbol{q}_j(\boldsymbol{y}) d\boldsymbol{y} =$$

$$\frac{1}{\sqrt{d_k}} \int_\Omega \boldsymbol{W}^P(\boldsymbol{x}, \boldsymbol{z}) \left( \boldsymbol{p}(\boldsymbol{z}) \boldsymbol{W}^Q \right) d\boldsymbol{z} \int_\Omega \left( \boldsymbol{q}(\boldsymbol{y}) \boldsymbol{W}^K \right)^\top \boldsymbol{q}_j(\boldsymbol{y}) d\boldsymbol{y}.$$

This reduces the computational complexity of the kernel-weighted integral in (7) from $O(N^2 d)$ to $O(N d^2)$. Additionally, we impose a convolutional structure on $\boldsymbol{W}^P$, and evaluate the first integral in the spectral domain:

$$\int_\Omega K[\boldsymbol{p}, \boldsymbol{q}; \boldsymbol{W}](\boldsymbol{x}, \boldsymbol{y}) \boldsymbol{q}_j(\boldsymbol{y}) d\boldsymbol{y} =$$

$$\frac{1}{\sqrt{d_k}} \mathcal{F}^{-1}(\boldsymbol{R} \cdot \mathcal{F}(\boldsymbol{p}\boldsymbol{W}^Q))(\boldsymbol{x}) \int_\Omega \left( \boldsymbol{q}(\boldsymbol{y}) \boldsymbol{W}^K \right)^\top \boldsymbol{q}_j(\boldsymbol{y}) d\boldsymbol{y},$$

where $\boldsymbol{R}$ is a learnable Fourier kernel that substitutes $\mathcal{F}(\boldsymbol{W}^P)$, $\boldsymbol{W}^Q$ and $\boldsymbol{W}^K$ are learnable matrices. Here, $\mathcal{F}$ and $\mathcal{F}^{-1}$ denote the Fourier transform and its inverse, respectively. This formulation reduces both parameter count and computational overhead. When taking $\boldsymbol{W}^P$ as a point-wise evaluation on the grid, it requires $O(N^2)$ memory, whereas learning $\boldsymbol{R}$ directly in Fourier space reduces memory footprint to $O(m)$, with $m \ll N$ being the number of retained Fourier modes. Consequently, the computational cost for the first integral is reduced from $O(N^2 d)$ to $O(N d \log N)$, leading to an overall complexity of $O(N d^2 + N d \log N)$ for NIPS.

### 3.4. NIPS: A Generalizable and Efficient PDE Solver

With the aforementioned modifications, the model is illustrated in Figure 1, along with the training algorithm summarized in Algorithm 1.

## 4. Experiments

We assess the performance of NIPS across a broad spectrum of physics modeling and discovery tasks, focusing on several key aspects. First, we highlight the advantages of using Fourier kernel with linear attention, and compare NIPS to various baseline models, including NAO with full LayerNorm (NAO-f), NAO with optimized normalization (NAO), NAO with full learnable square projection (NAO-$W^p$), and a NO with a convolution-based attention mechanism (AFNO (Guibas et al., 2021)). Our evaluation emphasizes generalizability, particularly zero-shot prediction performance when modeling new physical systems with unseen governing equations, as well as performance across different resolutions. Additionally, we examine the data efficiency-accuracy trade-off in inverse PDE learning tasks and assess the interpretability of the learned kernels. All experiments are optimized using the Adam optimizer. For fair comparison, we tune hyperparameters—including learning rates, decay rates, and regularization parameters—to minimize training loss. Experiments are conducted on a single NVIDIA A100 GPU with 40 GB of memory. Additional details on data generation and implementation can be found in the Appendix.

### 4.1. Darcy Flow

We begin by examining NIPS's ability to simultaneously learn both forward and inverse solution surrogates on the 2D Darcy flow benchmark. Specifically, we consider the modeling of 2D subsurface flows through a porous medium characterized by a heterogeneous permeability field. Following the setup in Yu et al. (2024), the high-fidelity synthetic simulation data for this problem are governed by the Darcy's flow. In this setting, the physical domain is $\Omega = [0, 1]^2$, $b(\boldsymbol{x})$ represents the permeability field, and the Darcy's equation

---

**Algorithm 1** The overall training algorithm and architecture of NIPS.

---

1: Given $\mathcal{D}_{tr} := \{\boldsymbol{f}_i^\eta(\boldsymbol{x}), \boldsymbol{u}_i^\eta(\boldsymbol{x})\}_{\eta=1,i=1}^{T^{tr},d}$ for training, $\mathcal{D}_{test} := \{\tilde{\boldsymbol{f}}_i^\eta(\boldsymbol{x}), \tilde{\boldsymbol{u}}_i^\eta(\boldsymbol{x})\}_{\eta=1,i=1}^{T^{test},d}$ for test sets. Each $\eta$ corresponds to a different PDE parameter field $\boldsymbol{b}^\eta$.

2: **Training Phase:**

3: **for** $ep = 1 : epoch_{max}$ **do**

4:     Update $\theta := \{\{\boldsymbol{R}_l, \boldsymbol{W}_l^Q, \boldsymbol{W}_l^K\}_{l=1}^L\}$, using Adam with respect to the optimization problem:

$$\theta^* = \underset{\theta}{\arg\min} \frac{1}{T_{tr}d} \sum_{\eta=1,i=1}^{T^{tr},d} \frac{\left|\left|\int_\Omega K[\mathbf{f}_{1:d}^\eta, \mathbf{u}_{1:d}^\eta; \theta](\boldsymbol{x}, \boldsymbol{y})\boldsymbol{f}_i^\eta(\boldsymbol{y})d\boldsymbol{y} - \boldsymbol{u}_i^\eta(\boldsymbol{x})\right|\right|_{L^2(\Omega)}}{||\boldsymbol{u}_i^\eta(\boldsymbol{x})||_{L^2(\Omega)}}, \tag{9}$$

where $K[\mathbf{f}_{1:d}^\eta, \mathbf{u}_{1:d}^\eta]$ is defined as:

$$\begin{aligned}
\boldsymbol{g}_j^{(0)}(\boldsymbol{x}) =& \boldsymbol{f}_j(\boldsymbol{x}), \ \boldsymbol{v}_j^{(0)}(\boldsymbol{x}) = \boldsymbol{u}_j(\boldsymbol{x}), \\
\boldsymbol{g}_j^{(l)}(\boldsymbol{x}) =& \boldsymbol{g}^{(l-1)}(\boldsymbol{x}) + \mathcal{N}\left(\frac{1}{\sqrt{d_k}}\mathcal{F}^{-1}(\boldsymbol{R}_l \cdot \mathcal{F}(\boldsymbol{g}^{(l-1)}\boldsymbol{W}_l^Q))(\boldsymbol{x}) \int_\Omega \left(\boldsymbol{g}^{(l-1)}(\boldsymbol{y})\boldsymbol{W}_l^K\right)^\top \boldsymbol{g}_j^{(l-1)}(\boldsymbol{y})d\boldsymbol{y}\right. \\
& \left. + \frac{1}{\sqrt{d_k}}\mathcal{F}^{-1}(\boldsymbol{R}_l \cdot \mathcal{F}(\boldsymbol{g}^{(l-1)}\boldsymbol{W}_l^Q))(\boldsymbol{x}) \int_\Omega \left(\boldsymbol{v}^{(l-1)}(\boldsymbol{y})\boldsymbol{W}_l^K\right)^\top \boldsymbol{v}_j^{(l-1)}(\boldsymbol{y})d\boldsymbol{y}\right), \\
\boldsymbol{v}_j^{(l)}(\boldsymbol{x}) =& \boldsymbol{v}^{(l-1)}(\boldsymbol{x}) + \mathcal{N}\left(\frac{1}{\sqrt{d_k}}\mathcal{F}^{-1}(\boldsymbol{R}_l \cdot \mathcal{F}(\boldsymbol{v}^{(l-1)}\boldsymbol{W}_l^Q))(\boldsymbol{x}) \int_\Omega \left(\boldsymbol{v}^{(l-1)}(\boldsymbol{y})\boldsymbol{W}_l^K\right)^\top \boldsymbol{v}_j^{(l-1)}(\boldsymbol{y})d\boldsymbol{y}\right. \\
& \left. + \frac{1}{\sqrt{d_k}}\mathcal{F}^{-1}(\boldsymbol{R}_l \cdot \mathcal{F}(\boldsymbol{v}^{(l-1)}\boldsymbol{W}_l^Q))(\boldsymbol{x}) \int_\Omega \left(\boldsymbol{g}^{(l-1)}(\boldsymbol{y})\boldsymbol{W}_l^K\right)^\top \boldsymbol{g}_j^{(l-1)}(\boldsymbol{y})d\boldsymbol{y}\right), \quad 1 \le l < L-1, \\
K[\mathbf{u}_{1:d}, \mathbf{f}_{1:d}; \theta](\boldsymbol{x}, \boldsymbol{y}) =& \frac{1}{\sqrt{d_k}}\mathcal{F}^{-1}(\boldsymbol{R}_L \cdot \mathcal{F}(\boldsymbol{g}^{(L-1)}\boldsymbol{W}_L^Q))(\boldsymbol{x}) \left(\boldsymbol{g}^{(L-1)}(\boldsymbol{y})\boldsymbol{W}_L^K\right)^\top \\
& + \frac{1}{\sqrt{d_k}}\mathcal{F}^{-1}(\boldsymbol{R}_L \cdot \mathcal{F}(\boldsymbol{v}^{(L-1)}\boldsymbol{W}_L^Q))(\boldsymbol{x}) \left(\boldsymbol{g}^{(L-1)}(\boldsymbol{y})\boldsymbol{W}_L^K\right)^\top. \tag{10}
\end{aligned}$$

5: **end for**

6: **Test Phase:**

7: Compute prediction error on the test dataset, as an evaluation for the forward PDE solver:

$$E_{forward}^{test} := \frac{1}{T_{test}d} \sum_{\eta=1,i=1}^{T^{test},d} \frac{\left|\left|\int_\Omega K[\tilde{\mathbf{f}}_{1:d}^\eta, \tilde{\mathbf{u}}_{1:d}^\eta; \theta^*](\boldsymbol{x}, \boldsymbol{y})\tilde{\boldsymbol{f}}_i^\eta(\boldsymbol{y})d\boldsymbol{y} - \tilde{\boldsymbol{u}}_i^\eta(\boldsymbol{x})\right|\right|_{L^2(\Omega)}}{||\tilde{\boldsymbol{u}}_i^\eta(\boldsymbol{x})||_{L^2(\Omega)}}.$$

8: If the ground-truth microstructure/kernel is available, compute kernel error on the test dataset, as an evaluation for the inverse PDE solver:

$$E_{inverse}^{test} := \frac{1}{T_{test}} \sum_{\eta=1}^{T^{test}} \frac{\left|\left|K[\tilde{\mathbf{f}}_{1:d}^\eta, \tilde{\mathbf{u}}_{1:d}^\eta; \theta^*](\boldsymbol{x}, \boldsymbol{y}) - K_{\tilde{\boldsymbol{b}}^\eta}(\boldsymbol{x}, \boldsymbol{y})\right|\right|_{L^2(\Omega\times\Omega)}}{\left|\left|K_{\tilde{\boldsymbol{b}}^\eta}(\boldsymbol{x}, \boldsymbol{y})\right|\right|_{L^2(\Omega\times\Omega)}}.$$

---

is given by:

$$-\nabla \cdot (b(\boldsymbol{x})\nabla p(\boldsymbol{x})) = g(\boldsymbol{x}), \quad \boldsymbol{x} \in \Omega,$$
$$p(\boldsymbol{x}) = 0, \quad \boldsymbol{x} \in \partial\Omega. \tag{11}$$

We aim to learn the solution operator of Darcy's equation and simultaneously compute the pressure field $p(\boldsymbol{x})$ and discover the underlying (hidden) permeability field $b(\boldsymbol{x})$.

**Ablation study.** We first conduct an ablation study on

NIPS by comparing its performance against its variants (i.e., NAO, NAO-f, NAO-$W^p$), using a fixed embedding dimension of $d = 50$, query-key projection dimension of $d_k = 40$, and a sequence length of $21 \times 21 = 441$. The results are reported in Table 1. All models are configured with two layers. Notably, NAO-f incorporates full LayerNorm in every layer, while NAO-$W^p$ employs a layerwise learnable projection, both of which significantly increase the number of trainable parameters. To isolate the effects of these modifications, we

keep the remaining architectural components unaltered. We first examine the impact of LayerNorm placement in NAO and NAO-f. In NAO, LayerNorm is applied across both the token and projection dimensions in the first layer, but only across the projection dimension in subsequent layers. In contrast, NAO-f normalizes across both dimensions in all layers. A comparison of test errors reveals that NAO outperforms NAO-f, achieving a 10.4% improvement in prediction accuracy while using 46.6% fewer trainable parameters. These results indicate that applying full LayerNorm across all layers is neither necessary nor beneficial, as it not only drastically increases the number of trainable parameters but also degrades model performance.

Next, we adopt the optimal LayerNorm placement and investigate whether incorporating layerwise learnable projections improves performance. Comparing NAO with NAO-$W^p$, we observe that the number of trainable parameters increases by more than 15 times, while accuracy deteriorates by 29.4%. This decline stems from the large square projection weights, whose size scales quadratically with the sequence length, explicitly computing and storing extensive pairwise token interactions. The substantial increase in trainable parameters also exacerbates overfitting, further degrading model performance. In contrast, NIPS achieves the highest accuracy while using the fewest trainable parameters among all models, outperforming the best NAO variant by 26.2% in accuracy. This improvement is attributed to NIPS's ability to implicitly model token dependencies by learning to interact with spectral modes, which encode a more concise representation of the data's intrinsic behavior. As a result, the model focuses more effectively on meaningful patterns while reducing redundancy. An additional advantage of learning spatial interactions in Fourier space is the substantial reduction in trainable parameters. Unlike NAO where weight size scales with the total number of spatial tokens, NIPS scales with the number of spectral modes considered, leading to significant computational savings. This efficiency gain is evident in per-epoch runtime comparisons: while NIPS slightly outperforms the linear version of NAO when the sequence length is relatively small, it accelerates training by a substantial 28.8% when the sequence length increases to $41 \times 41 = 1681$, accompanied by a 23.5% improvement in predictive accuracy.

**Comparison with additional baselines.** We then compare the performance of NIPS against an additional baseline, particularly a NO with a convolution-based attention mechanism, i.e., AFNO. To ensure a fair comparison, we maintain a comparable number of trainable parameters across different layer configurations and input sequence lengths. While AFNO's runtime is similar to that of NIPS, it fails to achieve zero-shot generalization to unseen PDE parameters, with test errors consistently exceeding 50% across all considered scenarios. Moreover, its performance deteriorates further as

*Table 1.* Test errors, per-epoch runtime, and number of trainable parameters for the Darcy flow problem, where bold numbers highlight the best method. The per-epoch runtimes for both the original quadratic attention and the re-formulated linear attention are reported and separated by "/" where applicable.

| Case | Model | #param | Per-epoch time (s) | Test error |
|---|---|---|---|---|
| 2 layers, 441 tokens $(21 \times 21)$ | NIPS | 98,396 | 1.6 | **2.28%** |
| | NAO | 101,134 | 5.4/1.7 | 3.09% |
| | NAO-f | 189,234 | 5.4/1.7 | 3.45% |
| | NAO-$W^p$ | 1,658,746 | 5.8/1.9 | 4.00% |
| | AFNO | 101,200 | 1.6 | 52.09% |
| 4 layers, 441 tokens $(21 \times 21)$ | NIPS | 108,692 | 2.5 | **1.03%** |
| | NAO | 109,494 | 7.8/2.5 | 1.45% |
| | AFNO | 121,600 | 1.9 | 52.84% |
| 4 layers, 1681 tokens $(41 \times 41)$ | NIPS | 366,932 | 8.4 | **1.14%** |
| | NAO | 357,494 | 58.8/11.8 | 1.49% |
| | AFNO | 364,000 | 24.5 | 98.70% |

model depth increases.

**Physics-informed data augmentation.** The above studies are conducted with a fixed number of random permutations, $n_{rand} = 100$, and a projection dimension of $d_k = 40$. Here, we investigate the impact of these two parameters. Random permutations in the embedding dimension help encode physical knowledge in the data by ensuring that features in the embedding space are not tied to a specific sequence order. This allows the model to disregard sequence effects in the embedding dimension and instead focus on learning spatial dependencies across tokens, which enhances generalization. Meanwhile, the projection dimension $d_k$ determines the level of information compression. As shown in Table 2, model performance improves monotonically with increasing $n_{rand}$ and $d_k$, eventually saturating when both parameters become sufficiently large.

**Interpretable physics discovery.** To demonstrate the physical interpretability of the learned kernel, we present in the first row of Figure 2 the ground-truth microstructure $b(\boldsymbol{x})$, a test loading field instance $g(\boldsymbol{x})$, and the corresponding solution $p(\boldsymbol{x})$. By summing the kernel strength along each row, one can discover the interaction strength of each material point $x$ with its neighbors. Since this strength correlates with the permeability field $b(x)$, the underlying microstructure can be recovered. The bottom row of Figure 2 shows the discovered microstructure for this test case. Due to the continuous nature of the learned kernel, the discovered microstructure appears smoothed, necessitating a thresholding step to distinguish the two-phase structure. The discovered microstructure (bottom-middle plot) matches well with the hidden ground-truth microstructure (top-left plot), except near the domain boundary. This discrepancy arises from the applied Dirichlet-type boundary condition ($p(x) = 0$ on

*Table 2.* Parametric studies on the effect of the number of random permutations and the size of $d_k$. A 2-layer NIPS with 441 tokens is adopted for demonstration.

| Case | $n_{rand}$ | $n_{train}$ | Test error |
|---|---|---|---|
| | 1 | 90 | 42.04% |
| | 2 | 180 | 28.98% |
| | 5 | 450 | 18.37% |
| $d_k = 40$ | 10 | 900 | 9.02% |
| | 25 | 2250 | 3.52% |
| | 50 | 4500 | 2.53% |
| | 100 | 9000 | 2.28% |

| Case | $d_k$ | $n_{train}$ | Test error |
|---|---|---|---|
| | 5 | 4500 | 14.67% |
| | 10 | 4500 | 6.93% |
| | 20 | 4500 | 4.03% |
| $n_{rand} = 50$ | 30 | 4500 | 3.01% |
| | 40 | 4500 | 2.53% |
| | 50 | 4500 | 2.45% |

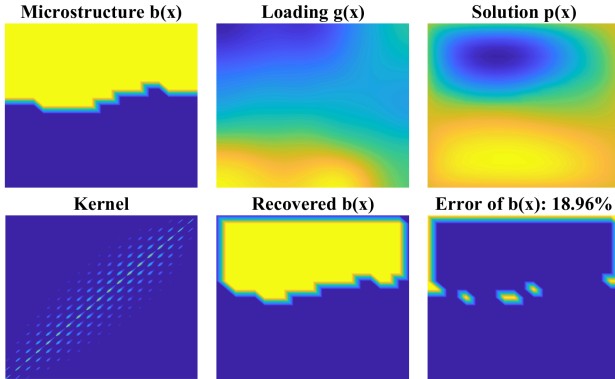

*Figure 2.* Interpretable microstructure discovery in experiment 1.

$\partial\Omega$) in all samples, which prevents the measurement pairs $(p(x), g(x))$ from containing information near $\partial\Omega$, making it impossible to identify the kernel at the boundaries.

With additional physics knowledge, one can quantitatively assess the interpretability provided by the recovered kernel. Specifically, the learned kernel should correspond to $K^{-1}$, where $K$ is the stiffness matrix of (11). Given this, the underlying permeability field $b(x)$ can be recovered by solving the following optimization problem: $B^* = \arg\min_B \sum_{i,j} \left\| K_B^{-1}[i,j] - K^{-1}[i,j] \right\|^2$, where $B = [b(x_1), \cdots, b(x_N)]$ denotes the pointwise value of $b$ at each grid point, and $K_B[i,j]$ represents the corresponding stiffness matrix obtained using a finite difference solver. In Table 3, we report the relative errors of both $K$ and $b$ on an $11 \times 11$ grid to quantitatively assess interpretability. Beyond the in-distribution (ID) setting where the microstructure $b$ is sampled from a Gaussian random field with covariance operator $(-\Delta + 5^2)^{-4}$ and the loading field $g$ from $(-\Delta + 5^2)^{-1}$, we also consider two out-of-distribution (OOD) scenarios: 1) ID microstructure $b$ with OOD loading field $g$, sampled

from $(-\Delta + 5^2)^{-4}$, and 2) OOD microstructure $b$ sampled from $(-\Delta + 5^2)^{-1}$ with ID loading field $g$. Intuitively, these OOD tasks are more challenging due to distributional shifts. Nevertheless, NIPS consistently achieves the lowest errors in both test performance and in the recovery of the kernel and microstructure across nearly all scenarios. Additionally, we consider a real-world data setting by introducing additional noise into the dataset. In particular, we perturb the training data with additive Gaussian noise: $\widetilde{g}(x) = g(x) + \epsilon(x), \epsilon \sim \mathcal{N}(0, \sigma^2)$, where $g(x)$ denotes the true source field and $\epsilon(x)$ represents zero-mean Gaussian noise with variance $\sigma^2$. Experiments are conducted for noise levels $\sigma = 0.01$ and $0.1$. As shown in Table 3, while NIPS's predictive accuracy naturally degrades with increasing noise, it remains robust overall, demonstrating resilience to observational perturbations.

### 4.2. Mechanical MNIST Benchmark

In this experiment, we study the learning of heterogeneous and nonlinear material responses using the Mechanical MNIST (MMNIST) dataset (Lejeune, 2020). MMNIST consists of 70,000 heterogeneous material specimens undergoing large deformations, each governed by a Neo-Hookean material model with a varying modulus derived from MNIST bitmap images. These material specimens are subjected to random 2D uniaxial extension, shear, equibiaxial extension, and confined compression loadings, with their corresponding material responses captured as prediction targets. The total number of tokens is thus $29 \times 29 \times 2 = 1682$, where "2" indicates the two in-plane directions.

In this and the following experiments, we directly compare NIPS with the best-performing baseline, NAO. Since the resolution is fixed at $29 \times 29$ in the MMNIST dataset, we evaluate model performance across different depths. As shown in Table 4, NIPS consistently outperforms NAO in both the 2-layer and 4-layer configurations, achieving performance gains of 59.2% and 78.9%, respectively. This underscores the advantages of implicitly learning token interactions in Fourier space, where NIPS captures intrinsic features (i.e., spectral modes) in data rather than relying on extensive discrete token dependencies.

### 4.3. Synthetic Tissue Learning

In the final experiment, we demonstrate NIPS's capability in learning synthetic tissues with highly organized structures, where collagen fiber arrangements vary spatially. Understanding the underlying hidden microstructural properties within the latent space of the complex, high-dimensional tissues is crucial, as they are inferred from experimental mechanical measurements. In this setting, we use loading data with a sequence length of $29 \times 29 \times 2 = 1682$ as input and aim to simultaneously learn a forward solution operator to predict the corresponding displacement field while

*Table 3.* Test errors, kernel errors, and microstructure errors for the Darcy flow problem, where bold numbers highlight the best method.

| Data Setting | Case | Model | #param | Test error | Kernel error | Microstructure error |
|---|---|---|---|---|---|---|
| ID | No noise | NIPS | 327,744 | **4.09%** | **9.24%** | 7.92% |
| | | NAO | 331,554 | 4.15% | 10.35% | **7.09%** |
| OOD, Scenario 1 | No noise | NIPS | 327,744 | **3.40%** | **10.63%** | **11.23%** |
| | | NAO | 331,554 | 6.86% | 14.46% | 20.37% |
| OOD, Scenario 2 | No noise | NIPS | 327,744 | **4.98%** | **9.68%** | **16.06%** |
| | | NAO | 331,554 | 5.57% | 10.94% | 20.30% |
| ID | Noise $\epsilon \sim \mathcal{N}(0, 0.01^2)$ | NIPS | 327,744 | 4.40% | 9.47% | 7.97% |
| OOD, Scenario 1 | Noise $\epsilon \sim \mathcal{N}(0, 0.01^2)$ | NIPS | 327,744 | 3.88% | 11.14% | 13.63% |
| OOD, Scenario 2 | Noise $\epsilon \sim \mathcal{N}(0, 0.01^2)$ | NIPS | 327,744 | 5.65% | 10.21% | 17.36% |
| ID | Noise $\epsilon \sim \mathcal{N}(0, 0.1^2)$ | NIPS | 327,744 | 9.98% | 21.77% | 14.84% |
| OOD, Scenario 1 | Noise $\epsilon \sim \mathcal{N}(0, 0.1^2)$ | NIPS | 327,744 | 3.84% | 21.08% | 19.87% |
| OOD, Scenario 2 | Noise $\epsilon \sim \mathcal{N}(0, 0.1^2)$ | NIPS | 327,744 | 11.67% | 22.77% | 26.54% |

*Table 4.* Test errors and number of trainable parameters for the MMNIST benchmark. Bold numbers highlight the best method.

| Case | Model | #param | Test error |
|---|---|---|---|
| 2 layers with | NIPS | 720,600 | **2.11%** |
| 1682 tokens | NAO | 722,748 | 5.17% |
| 4 layers with | NIPS | 768,600 | **1.11%** |
| 1682 tokens | NAO | 763,548 | 5.27% |

*Table 5.* Test errors and number of trainable parameters for experiment 3, where bold numbers highlight the best method.

| Case | Model | #param | Test error |
|---|---|---|---|
| 4 layers with | NIPS | 768,600 | 4.95% |
| 1682 tokens | NIPS-mlp | 2,183,582 | **4.52%** |
| | NAO | 763,548 | 5.62% |

uncovering the underlying microstructures in the tissues.

We present our experimental results in Table 5, comparing the 4-layer NIPS with NAO of the same depth. This problem exhibits significant nonlinearity due to the inherent complexity of the tissue microstructure. Nevertheless, NIPS successfully learns an accurate solution operator, achieving a test error within 5% and outperforming NAO by 11.9%. Notably, NIPS's predictive accuracy can be further improved by passing the $V$ vector through an MLP before its matrix multiplication with the $K$ vector in the final layer, enabling nonlinear mixing of the loading data. To illustrate this idea, we introduce a simple two-layer MLP with skip connection and report the results in Table 5 as NIPS-mlp. This modification improves accuracy by an additional 7.7%, with further gains possible through careful MLP design. However, it also nearly triples the number of trainable parameters. Given the

somewhat marginal performance improvement relative to the increased model complexity, we opt not to pursue this direction further.

## 5. Conclusion

We present NIPS, a novel attention-based neural operator architecture for simultaneous forward PDE solving and governing physics discovery. NIPS integrates the attention mechanism with a learnable kernel network for channel-independent convolution in Fourier space, effectively eliminating the need to compute and store large pairwise interactions and amortizing the cost of handling spatial interactions into the Fourier transform. Through reformulation in a linear attention perspective, NIPS harmonizes Fourier insights with attention for scalable and interpretable physics discovery. Our zero-shot learning experiments demonstrate that NIPS generalizes effectively across both forward and inverse PDE learning tasks, thereby enhancing physical interpretability and enabling more robust generalization across diverse physical systems.

**Limitations:** As discussed in NAO, the core idea of attention-based operators is to extract prior knowledge across multiple PDE tasks, in the form of an identifiable kernel space. This approach assumes that the target kernel shares structural similarities with those seen during training. Consequently, performance may degrade when the target kernel significantly deviates from the training distribution. For example, if the operator is trained solely on diffusion problems, where all kernels (i.e., stiffness matrices) are symmetric, the learned prior will inherently favor symmetric structures. As a result, NIPS may struggle to predict the stiffness matrix for systems with non-symmetric kernels, such as those arising in advection-dominated problems.

## Acknowledgments

The authors would like to thank Yiming Fan for the helpful discussion. Y. Yu would like to acknowledge support by the National Science Foundation (NSF) under award DMS-2427915, the Department of the Air Force under award FA9550-22-1-0197, and the National Institute of Health under award 1R01GM157589-01. Portions of this research were conducted on Lehigh University's Research Computing infrastructure partially supported by NSF Award 2019035.

## Impact Statement

This paper presents work whose goal is to advance the field of machine learning. There are many potential societal consequences of our work, none of which we feel must be specifically highlighted here.

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

## A. Data Generation and Additional Results

### A.1. Experiment 1 - Darcy Flow

We generate synthetic data based on Darcy flow in a square domain $\Omega = [0, 1]^2$ subjected to Dirichlet boundary conditions. The problem is described by the equation: $-\nabla(b(x)\nabla p(x)) = g(x)$ with $p(x) = 0$ on all boundaries. This equation models diffusion in heterogeneous fields, such as subsurface water flow in porous media, where the heterogeneity is represented by the location-dependent conductivity $b(x)$. Here, $p(x)$ is the source term, and $g(x)$ is the hydraulic height (the solution). For each sample, we solve the equation on both $21 \times 21$ and $41 \times 41$ grids using an in-house finite difference code. We consider 100 random microstructures, each consisting of two distinct phases. In particular, we sample from a Gaussian random field with zero mean and covariance operator $(-\Delta + 5^2)^{-4}$ for the in-distribution (ID) scenario, and $(-\Delta + 5^2)^{-1}$ for the out-of-distribution (OOD) scenario. For each sample $\xi$, we divide the square domain into two subdomains, with conductivities $b(x) = 12$ when $\xi(x) < 0$ and $b(x) = 3$ when $\xi(x) \geq 0$. Additionally, we generate 100 distinct source functions $g(x)$ using a Gaussian random field generator with zero mean and covariance operator $(-\Delta + 5^2)^{-1}$ for the ID case, and $(-\Delta + 5^2)^{-4}$ for the OOD case. For each microstructure, we solve the Darcy problem with all 100 source terms, yielding a dataset of $N = 10,000$ function pairs in the form of $\{p_i(x_j), g_i(x_j)\}_{i=1}^N$, where $j = 1, 2, \cdots, 441$ for the $21 \times 21$ mesh and $j = 1, 2, \cdots, 1681$ for the $41 \times 41$ mesh, with $x_j$ denoting the discretization points on the square domain.

In operator learning, we note that the permutation of function pairs in each sample should not alter the learned kernel, i.e.,

$$K[\mathbf{u}_{1:d}, \mathbf{f}_{1:d}] = K[\mathbf{u}_{\sigma(1:d)}, \mathbf{f}_{\sigma(1:d)}], \tag{12}$$

where $\sigma$ is the permutation operator. To address this, we augment the training dataset by permuting the function pairs in each task. Specifically, with 100 microstructures (tasks) and 100 function pairs per task, we randomly permute the function pairs and take 100 different permutations per task. This process generates a total of 10,000 samples (9,000 for training and 1,000 for testing) in the form of $\{\boldsymbol{u}_{1:100}^\eta, \boldsymbol{f}_{1:100}^\eta\}_{\eta=1}^{10,000}$.

We demonstrate the scalability of NIPS using larger discretizations (e.g., $121 \times 121$ with 14,641 tokens in total), and compare its per-epoch runtime and peak GPU memory usage with the best-performing baseline, NAO. The results are summarized in Table 6, where "x" indicates that the method exceeds memory limits on a single NVIDIA A100 GPU with 40 GB memory due to the explicit computation of spatial token interactions. Compared to the quadratic NAO that fails at 3.7k tokens and the linear NAO that fails at 6.5k tokens, NIPS can easily scale up to 15k tokens on a single A100 GPU. Note that the analysis is performed on a single GPU. To further accelerate training, one can utilize multiple GPUs and leverage distributed training frameworks, such as PyTorch's Distributed Data Parallel (DDP) module.

*Table 6.* Scalability demonstration via per-epoch runtime and peak memory footprint for the Darcy flow problem. All models are 4-layer. The per-epoch runtimes for both the original quadratic attention and the re-formulated linear attention are reported and separated by "/". "x" denotes exceeding memory limits on a single NVIDIA A100 GPU with 40 GB memory.

|  | # tokens | 441 | 1,681 | 3,721 | 6,561 | 10,201 | 14,641 |
|---|---|---|---|---|---|---|---|
| Per-epoch runtime (s) | NIPS | 2.5 | 8.4 | 17.1 | 25.1 | 45.6 | 65.66 |
|  | NAO | 7.8/2.5 | 58.8/11.8 | x/20.9 | x/x | x/x | x/x |
| Peak memory usage (GB) | NIPS | 0.46 | 1.68 | 3.68 | 6.46 | 10.04 | 14.39 |
|  | NAO | 2.47/0.66 | 32.61/5.69 | x/25.53 | x/x | x/x | x/x |

We repeat the experiments three times with randomly selected seeds in the first case of experiment 1, using NIPS and the major baseline models including NAO and AFNO. The results are reported in Table 7. Consistent with the previous findings, NIPS outperforms the best baseline by 28.9%.

*Table 7.* Test errors and number of trainable parameters for the Darcy flow problem. Bold numbers highlight the best method.

| Model | NIPS | NAO | AFNO |
|---|---|---|---|
| Test error | **2.31%±0.03%** | 3.25%±0.14% | 52.92%±0.72% |

## A.2. Experiment 2 - Mechanical MNIST benchmark

Mechanical MNIST is a benchmark dataset of heterogeneous materials undergoing large deformation, modeled by Neo-Hookean material with a varying modulus derived from MNIST bitmap images (Lejeune, 2020). It contains 70,000 material specimens, each governed by the Neo-Hookean model with a modulus varying according to the MNIST images. Figure 3 illustrates samples from the MMNIST dataset, including the underlying microstructure, randomly selected loading fields, and the corresponding displacement fields in the two in-plane directions.

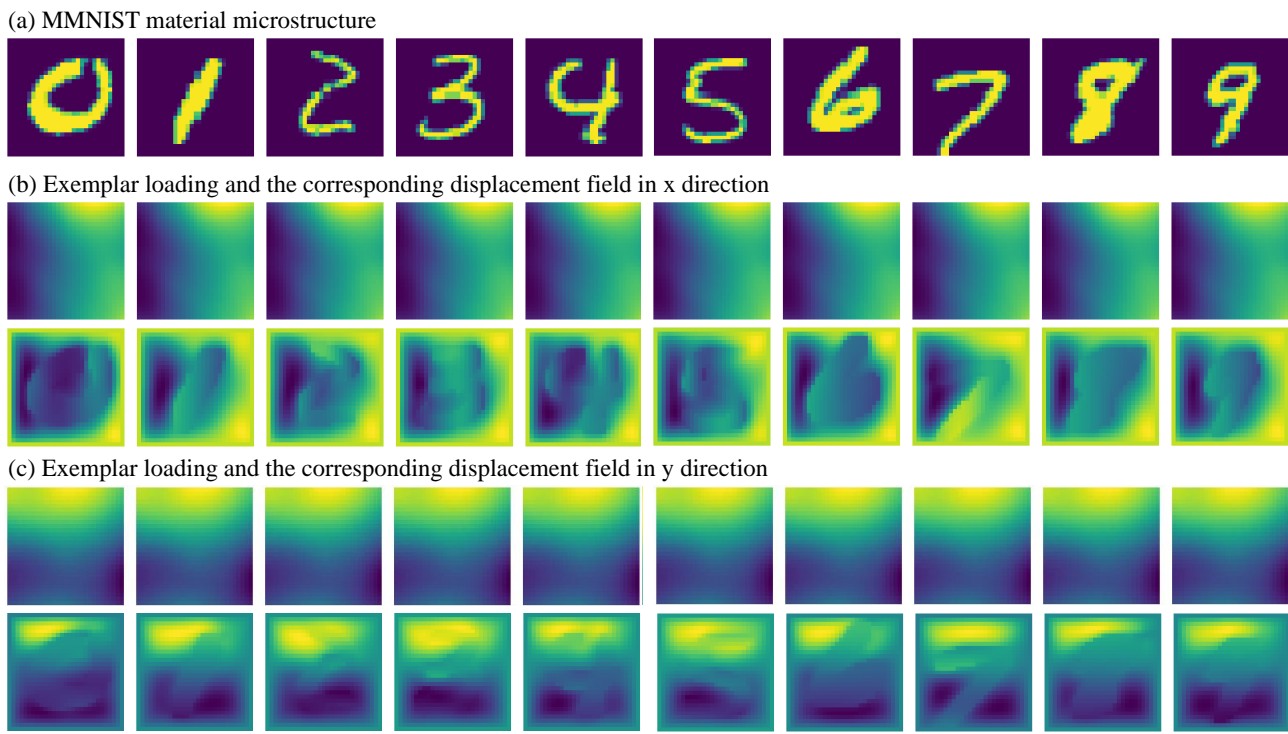

*Figure 3*. Illustration of exemplar MMNIST samples in experiment 2. (a): material parameter field corresponding to different $\boldsymbol{b}$. (b): displacement fields (second row) $\boldsymbol{u}_x$ corresponding to the same loading field (first row) $\boldsymbol{f}_x$. (c): displacement fields (second row) $\boldsymbol{u}_y$ corresponding to the same loading field (first row) $\boldsymbol{f}_y$.

## A.3. Experiment 3 - Synthetic Tissue Learning

We generate synthetic tissue data by sampling the fiber orientation distribution for each sample, which consists of two segments with orientations $\alpha_1$ and $\alpha_2$ on each side, respectively, separated by a line passing through the center of a square domain. The values of $\alpha_1$ and $\alpha_2$ are independently sampled from a uniform distribution over $\mathcal{U}[0, 2\pi]$, and the centerline's rotation is sampled from $\mathcal{U}[0, \pi]$. We generate 500 material sets, each containing 100 loading/displacement pairs, and divide these into training and test sets with a 450/50 split. The loading in this dataset is taken as the body load, $\boldsymbol{f}(\boldsymbol{x})$. Each instance is generated as the restriction of a 2D random field, $\phi(\boldsymbol{x}) = \mathcal{F}^{-1}(\gamma^{1/2}\mathcal{F}(\Gamma))(\boldsymbol{x})$, where $\Gamma(\boldsymbol{x})$ is a Gaussian white noise random field on $\mathbb{R}^2$, $\gamma = (w_1^2 + w_2^2)^{-\frac{5}{4}}$ represents a correlation function, and $w_1$ and $w_2$ are the wave numbers on $x$ and $y$ directions, respectively. The operators $\mathcal{F}$ and $\mathcal{F}^{-1}$ denote the Fourier transform and its inverse, respectively. This random field is expected to have a zero mean and covariance operator $C = (-\Delta)^{-2.5}$, with $\Delta$ being the Laplacian under periodic boundary conditions on $[0, 2]^2$, and we then restrict it to $\Omega$. For details on Gaussian random field sample generation, we refer readers to Lang & Potthoff (2011). Finally, for each sampled loading field $\boldsymbol{f}_i(\boldsymbol{x})$ and microstructure field $\boldsymbol{b}(\boldsymbol{x})$, we solve for the displacement field $\boldsymbol{u}_i(\boldsymbol{x})$ on the entire domain.

# B. Implementation Details and Further Discussion on Baselines

The configuration of each baseline is detailed below, where the parameter choice of each model is selected by tuning the number of layers and the width (channel dimension), ensuring that the total number of parameters remains within the same order of magnitude. We follow the two key principles in McGreivy & Hakim (2024) for fair comparisons: (1) evaluating models at either equal accuracy or equal runtime, and (2) comparing against an efficient numerical method. First, we emphasize that our comparisons focus on state-of-the-art (SOTA) machine learning methods rather than standard numerical solvers used in data generation. Specifically, we evaluate our proposed model, NIPS, against NAO, its variants, and AFNO, which are established baseline models in the field. To ensure fairness (rule 1), we maintain a comparable total number of trainable parameters across models, as large discrepancies could lead to misleading conclusions. Within this constraint, we assess (1) per-epoch runtime to measure computational efficiency and (2) test errors to evaluate prediction accuracy. Regarding rule 2, we select SOTA neural operator models as baselines to demonstrate the advantages of NIPS in accuracy and efficiency. These models represent the strongest available baselines for learning-based PDE solvers. Therefore, our experimental setup adheres to the principles outlined in McGreivy & Hakim (2024).

- NAO: We use a 4-layer NAO model with LayerNorm applied across both the token and projection dimensions in the first layer, but only across the projection dimension in subsequent layers. Both the original quadratic attention and the reformulated linear attention are tested. We parameterize the kernel network $W^{P,u}$ and $W^{P,f}$ with a 3-layer MLP with hidden dimensions $(32, 64)$ and LeakyReLU activation functions.

- NAO-f: The NAO-f model follows the same configuration as NAO, except that LayerNorm is applied across both the token and projection dimensions in all layers.

- NAO-$W^p$: The NAO-$W^p$ model follows the same configuration as NAO, except that a learnable projection weight of size $[2 \times n_{\text{tokens}}, 2 \times n_{\text{tokens}}]$ is added to each iterative layer.

- AFNO: We closely follow the setup in Guibas et al. (2021) and stack 4 AFNO layers together to form the final AFNO model. For each AFNO layer, the number of blocks is set to 1, and the channel dimension size is set to the embedding size with the hidden size factor equal to 3. All other parameters such as the sparsity threshold and the hard thresholding fraction are set to the default values of 0.01 and 1, respectively.

Note that NIPS is conceptually related to the Performer (Choromanski et al., 2020), which introduces kernel-based approximations for efficient self-attention. Performer replaces the standard softmax attention with positive orthogonal random features, enabling linear scaling with respect to sequence length. While NIPS similarly aims to mitigate the quadratic complexity of self-attention, NIPS achieves this through a Fourier-domain reformulation that leverages convolutional structure. Unlike Performer's randomized feature mappings, NIPS deterministically decomposes attention using spectral representations, making it particularly well-suited for physics-based PDE learning. Additionally, NIPS is also related to Graph Neural Operators (GNOs) (Li et al., 2020a), which provide a graph-based formulation for learning nonlocal interactions in operator learning tasks. GNOs use graph message passing to propagate information over irregular domains, effectively capturing spatial dependencies. In contrast, NIPS operates in the Fourier domain, leveraging spectral representations to encode long-range dependencies in a continuous and structured manner. While graph-based architectures such as GNOs and NAO can learn operators on discretized graph structures, our Fourier-based formulation provides a leveraged efficiency on structured grids.

