# OpenReview forum: "Neural Interpretable PDEs: Harmonizing Fourier Insights with Attention for Scalable and Interpretable Physics Discovery"
_ICML.cc/2025/Conference — ICML 2025 poster_

### Official Review · Reviewer_JrVi · 2025-03-12

**Overall Recommendation:** 2

**Summary:**

This paper introduces Neural Interpretable PDEs (NIPS), a novel neural operator architecture that enhances both predictive accuracy and computational efficiency in modeling complex physical systems. NIPS builds upon Nonlocal Attention Operators (NAO) by employing a linear attention mechanism combined with a learnable kernel network that functions as a channel-independent convolution in Fourier space. This design eliminates the need to explicitly compute and store large pairwise interactions, effectively amortizing the cost of spatial interactions through the Fourier transform.

**Claims And Evidence:**

The claims made in this submission are generally well-supported by empirical evidence and theoretical analysis. The authors provide comprehensive quantitative results across three distinct experimental setups (Darcy flow, MMNIST, and synthetic tissue learning), consistently demonstrating NIPS's superior performance compared to baseline models.

**Essential References Not Discussed:**

While the authors cite linear attention work by Cao (2021), they do not discuss Performer (Choromanski et al., 2020, NeurIPS) which introduced kernel-based approximations for efficient attention that are conceptually related to their Fourier-domain approach. Second, the paper could benefit from citing Graph Neural Operators (GNOs) by Li et al. (2020, NeurIPS), as these provide an alternative perspective on handling non-local interactions in operator learning that would help contextualize NIPS's approach.

**Experimental Designs Or Analyses:**

The experimental designs in this paper are generally sound and well-executed. The Darcy flow experiments (Section 4.1) use appropriate synthetic data generation procedures with clear controls across model variants while maintaining comparable parameter counts.

The parametric studies in Table 2 systematically explore the effects of random permutations and projection dimensions with sufficient sampling to establish trends. One limitation is that statistical significance of the performance differences is not explicitly established through multiple runs with different random seeds.

**Methods And Evaluation Criteria:**

The proposed methods and evaluation criteria in the paper are well-aligned with the challenges of learning interpretable operators for physical systems. The NIPS architecture logically addresses the limitations of existing approaches by combining Fourier techniques with linear attention, which is appropriate for handling PDE operators efficiently while maintaining interpretability.

**Other Comments Or Suggestions:**

NA

**Other Strengths And Weaknesses:**

Pros:

The harmonization of Fourier insights with attention mechanisms represents a creative combination of existing ideas that yields tangible performance improvements. The dual capability of NIPS to solve both forward and inverse problems within a unified framework is especially significant, as it addresses a fundamental challenge in computational physics.

Cons:

While empirical results are strong, a more rigorous mathematical foundation would strengthen the paper's contribution. Additionally, the experimental evaluation could benefit from a more detailed analysis of failure cases or boundary conditions where the method might underperform. The paper would also be strengthened by exploring more diverse or complex physical systems beyond the three test cases presented, particularly systems with higher dimensionality or stronger nonlinearities.

**Questions For Authors:**

1. Your experiments show AFNO consistently achieving error rates exceeding 50% across all scenarios, which seems unusually poor for an established method. Could you clarify whether this reflects a fundamental limitation of AFNO for these specific tasks, or if there might be implementation factors affecting its performance?

2. While you demonstrate strong zero-shot generalization capabilities within similar PDE systems, how would NIPS perform when transferring to fundamentally different classes of PDEs? For instance, could a model trained on diffusion equations generalize to wave equations or advection-dominated systems?

**Relation To Broader Scientific Literature:**

The key contributions of this paper build upon several important strands of research in the computational physics and machine learning literature. NIPS integrates two significant developments in neural operators: the Fourier Neural Operator (FNO) framework introduced by Li et al. (2020c), which leverages spectral methods for efficient computation, and the Nonlocal Attention Operators (NAO) recently proposed by Yu et al. (2024), which focus on interpretability through attention mechanisms.

**Theoretical Claims:**

The paper does not present formal mathematical proofs in the traditional sense with explicitly stated theorems and detailed proof steps

---

> ### Author Rebuttal · Authors · 2025-04-01
>
> We thank the reviewer for the insightful comments. Our response:
>
> **Multiple runs with different random seeds**: We repeat the experiments three times with randomly selected seeds in the first case of experiment 1, using NIPS and the major baseline models including NAO and AFNO. The results are reported in the table below. Consistent with the findings in the original manuscript, NIPS outperforms the best baseline by 28.9\%.
>
> Table 1: Test errors and number of trainable parameters for the Darcy flow problem. Bold numbers highlight the best method.
> |Model | NIPS | NAO | AFNO |
> | :------------- | :-----------: | :-----------: | :-----------: |
> |Test error | **2.31\%**$\pm$**0.03\%** | 3.25\%$\pm$0.14\% | 52.92\%$\pm$0.72\% |
>
> **Additional discussion on Performer and GNO**: We appreciate the suggestion and agree that additional discussion on these works will help contextualize NIPS. NIPS is conceptually related to Performer, which introduces kernel-based approximations for efficient self-attention by replacing softmax attention with positive orthogonal random features, achieving linear scaling with sequence length. While both aim to reduce the quadratic complexity of self-attention, NIPS employs a Fourier-domain reformulation that leverages convolutional structure. Unlike Performer’s randomized feature mappings, NIPS deterministically decomposes attention using spectral representations, making it well-suited for physics-based PDE learning. Our work is also related to GNOs, which use graph message passing to model nonlocal interactions in operator learning tasks. While GNOs capture spatial dependencies over irregular domains, NIPS operates in the Fourier domain, leveraging spectral representations to encode long-range dependencies in a structured manner. Whereas graph-based architectures like GNOs and NAO learn operators on discretized graph structures, our Fourier-based approach is particularly efficient on structured grids. We will incorporate this discussion in the revised manuscript.
>
> **More rigorous mathematical foundation**: We appreciate the suggestion. NIPS inherits resolution invariance and improved identifiability from NAO, and we will add this discussion to the manuscript. While further mathematical analysis is an interesting future direction, our current focus is on developing a novel architecture that enhances predictive accuracy and computational efficiency for both forward and inverse PDE problems.
>
> **More detailed analysis of challenging and failure cases**: To explore where NIPS might fail, we consider two out-of-distribution scenarios. The training dataset uses a microstructure $b$ generated by a Gaussian random field with covariance operator $(-\Delta + 5^2)^{-4}$ and a loading field $g$ generated by $(-\Delta + 5^2)^{-1}$. The out-of-distribution scenarios are: 1) in-distribution (ID) microstructure $b$ and out-of-distribution (OOD) $g$ using $(-\Delta + 5^2)^{-4}$; 2) OOD microstructure $b$ using $(-\Delta + 5^2)^{-1}$ and ID $g$. Results below show that both NIPS and NAO perform well across these scenarios in terms of forward operator errors and kernel errors, but microstructure errors increase, especially in scenario 2, where the out-of-distribution microstructure is rougher and more detailed. This complexity makes it harder to recover the ground truth, particularly in an unsupervised learning context.
>
> Table 2: Test, kernel, and microstructure errors for Darcy flow. Bold numbers highlight the best method.
> |Data Setting | Case | Model | \#param | Test error | Kernel error | Microstructure error |
> | :------------- | :-----------: | :-----------: | :-----------: | :-----------: | :------------- | :-----------: |
> |ID | No noise | NIPS | 327,744 | **4.09\%** | **9.24\%** | 7.92\% |
> | | | NAO | 331,554 | 4.15\% | 10.35\% | **7.09\%** |
> |OOD, Scenario 1 | No noise | NIPS | 327,744 | **3.40\%** | **10.63\%** | **11.23\%** |
> | | | NAO |331,554  | 6.86\% | 14.46\% | 20.37\% |
> |OOD, Scenario 2|No noise | NIPS | 327,744 | **4.98\%** | **9.68\%** | **16.06\%** |
> | | | NAO |331,554  | 5.57\% | 10.94\% | 20.30\% |
>
> **Explanation on AFNO performance**: AFNO performs poorly because it is designed to learn solutions for a single PDE system and cannot handle inputs from multiple systems. In contrast, both NAO and NIPS leverage global prior knowledge from training data of multiple systems, allowing them to generalize across unseen system states.
>
> **Transferring NIPS to fundamentally different PDEs**: As mentioned in NAO, attention operators extract prior knowledge in the form of identifiable kernel spaces from multiple PDE tasks. This approach may not work well if the kernel for a new problem differs significantly from those in the training data. For example, if NIPS is trained on diffusion problems with symmetric kernels (stiffness matrices), it will struggle to predict stiffness matrices for non-symmetric systems, like those in advection problems. We will include this discussion in the revised paper.

---

### Official Review · Reviewer_yXYz · 2025-03-14

**Overall Recommendation:** 3

**Summary:**

This article introduces an operator learning method that computes the solution to a PDE with a data-dependent kernel called NIPS. The kernel comes from a linear attention mechanism performs in Fourier space, which improves the efficiency of the method. The method can also be used for the inverse problem to discover the governing parameters of a PDE given the solution. This method is tested Darcy flow, mechanical MNIST, and synthetic tissue models. The methods is claimed to be accurate, efficient, and interpretable.

**Claims And Evidence:**

The claim of efficiency is not substantiated clearly. The computational domains of the computer experiments remain fairly small. All discretization are under 2000 points which does not allow for large computational domains. Larger domains need to be studied to illustrate the efficiency and scalability of the method.

**Essential References Not Discussed:**

NA

**Experimental Designs Or Analyses:**

See concern above.

**Methods And Evaluation Criteria:**

The chosen applications are also weak baselines [1]. More computational experiments are needed to benchmark the method.

McGreivy, Nick, and Ammar Hakim. "Weak baselines and reporting biases lead to overoptimism in machine learning for fluid-related partial differential equations." Nature Machine Intelligence 6.10 (2024): 1256-1269.

**Other Comments Or Suggestions:**

NA

**Other Strengths And Weaknesses:**

NA

**Questions For Authors:**

NA

**Relation To Broader Scientific Literature:**

This work is an extension to Nonlocal Attention Operators with linear attention in the Fourier domain.

**Theoretical Claims:**

This is an empirical study.

---

> ### Author Rebuttal · Authors · 2025-04-01
>
> We thank the reviewer for the insightful comments and questions. Our response:
>
> **Larger discretization**: We demonstrate the scalability of NIPS using larger discretizations  (e.g., $121\times121$ with 14,641 tokens in total), and compare its per-epoch runtime and peak GPU memory usage with the best-performing baseline, NAO. The results are summarized in the table below, where ``x'' indicates that the method exceeds memory limits on a single NVIDIA A100 GPU with 40 GB memory due to the explicit computation of spatial token interactions. Compared to the quadratic NAO that fails at 3.7k tokens and the linear NAO that fails at 6.5k tokens, NIPS can easily scale up to 15k tokens on a single A100 GPU. Note that the analysis is performed on a single GPU. To further accelerate training, one can utilize multiple GPUs and leverage distributed training frameworks, such as PyTorch’s Distributed Data Parallel (DDP) module, which will be an interesting direction for future work.
>
> Table 1: Scalability demonstration via per-epoch runtime and peak memory footprint for the Darcy flow problem. All models are 4-layer. The per-epoch runtimes for both the original quadratic attention and the re-formulated linear attention are reported and separated by /. ``x'' denotes exceeding memory limits on a single NVIDIA A100 GPU with 40 GB memory.
> |                         | \# tokens |     441  | 1,681 | 3,721  | 6,561  | 10,201 | 14,641 |
> |-------------------------|-----------:|--------:|--------:|--------:|--------:|--------:|--------:|
> | **Per-epoch runtime (s)** | **NIPS**  |       2.5    | 8.4    | 17.1   | 25.1   | 45.6   | 65.66  |
> |                         | **NAO**   | 7.8/2.5| 58.8/11.8| x/20.9| x/x    | x/x    | x/x    |
> | **Peak memory usage (GB)** | **NIPS**  | 0.46   | 1.68   | 3.68   | 6.46   | 10.04  | 14.39  |
> |                         | **NAO**   | 2.47/0.66 | 32.61/5.69 | x/25.53 | x/x   | x/x    | x/x    |
>
> **Weak-baseline applications**: Thank you for pointing us to this insightful work. The referenced paper highlights two key principles for fair comparisons: (1) evaluating models at either equal accuracy or equal runtime, and (2) comparing against an efficient numerical method. First, we clarify that our comparisons focus on state-of-the-art (SOTA) machine learning methods rather than standard numerical solvers used in data generation. Specifically, we evaluate our proposed model, NIPS, against NAO, its variants, and AFNO, which are established baseline models in the field. To ensure fairness (rule 1), we maintain a comparable total number of trainable parameters across models, as large discrepancies could lead to misleading conclusions. Within this constraint, we assess (1) per-epoch runtime to measure computational efficiency and (2) test errors to evaluate prediction accuracy. Regarding rule 2, we select SOTA neural operator models as baselines to demonstrate the advantages of NIPS in accuracy and efficiency. These models represent the strongest available baselines for learning-based PDE solvers. Therefore, our experimental setup adheres to the principles outlined in the referenced paper. We will incorporate this discussion into our revised manuscript to further clarify our evaluation methodology.

---

### Official Review · Reviewer_JpKF · 2025-03-14

**Overall Recommendation:** 4

**Summary:**

The paper introduces Neural Interpretable PDEs (NIPS), an attention-based neural operator architecture for solving forward and inverse PDE problems. NIPS utilizes a learnable kernel network to optimize efficiency in Fourier transform interactions by mitigating the computation and storage of the pairwise interactions. Improved upon Nonlocal Attention Operators (NAO), the author incorporates linear attention to improve the scalability and interpretability of large physical systems learning. The authors demonstrate that NIPS surpasses NAO and other baselines across multiple physics modeling benchmarks.

**Claims And Evidence:**

The author claims efficiency in Fourier Transform interaction, interpretability and scalability based on the improved attention structure, and generalizability on zero-shot unseen physical systems. The numerical results in Table 1 provide evidence that NIPS has lower test errors and faster training times in the Darcy Flow problem compared to the baselines. Interpretability is primarily demonstrated through visualizations in the Darcy Flow section. While these visualizations provide a qualitative explanation, it is suggested to extend them to other tested datasets or incorporate formalized metrics, as interpretability is a relatively subjective term.

**Essential References Not Discussed:**

The author discusses the majority of essential references.

**Experimental Designs Or Analyses:**

The experiments confirm NIPS' capability in forward and inverse PDE solving as well as zero-shot generalization. Some vital baseline for this method, such as NAO and NAO-relatives, are included. Additionally, multiple cases of token and layer numbers are discussed. One suggestion is to include real-world PDE data to further demonstrate NIPS' practical applicability, such as airfoil flow and climate data. They could provide additional insights into how well NIPS generalizes on noisy data, which is beyond synthetic PDE problems.

**Methods And Evaluation Criteria:**

The benchmark datasets utilized are 2D Darcy Flow, Mechanical MNIST, and Synthetic Tissue Learning. Unlimited to fluid dynamics, the dataset also incorporates a material and biomechanics dataset. The evaluation metric utilized is mainly relative mean squared error (rMSE), which is a standardized metric in evaluating prediction performances.

**Other Comments Or Suggestions:**

Suggestions are mentioned in the earlier sections.

**Other Strengths And Weaknesses:**

Clarity: The paper flows smoothly, making readers easy to follow.

**Questions For Authors:**

Question 1: How would interpretability vary across different zero-shot conditions?

**Relation To Broader Scientific Literature:**

This paper proposes a useful method that is closely related to multiple facets in PDE learning, including Neural Operators (NO) and Fourier Neural Operators (FNO), its prior work Nonlocal Attention Operators (NAO). It is also related to both forward and inverse PDE solving in efficiency and effectiveness considerations.

**Theoretical Claims:**

The theoretical claims in Section 3.2 and Section 3.3 introduce the expression of attention mechanism-based kernel map and the reformulation of Fourier-domain kernel to enhance the efficiency. Specifically, the reformulation and the assessment of Big O in Section 3.3 looks correct to me.

---

> ### Author Rebuttal · Authors · 2025-04-01
>
> We thank the reviewer for the insightful comments . Our response:
>
> **Formalized interpretability metric**: We thank the reviewer's valuable suggestion. A quantitative evaluation of the interpretability can be provided by the recovered kernel. Taking the Darcy flow example for instance, the discovered kernel should correspond to the stiffness matrix $K$, and the underlying permeability field $b(x)$ can be obtained by solving an optimization problem: $B^*=argmin_B \sum_{i,j}||K_B[i,j]-K[i,j]||^2$. Here $B=[b(x_1),\cdots,b(x_N)]$ denotes the pointwise value of $b$ at each grid point, and $K_B[i,j]$ is the corresponding stiffness matrix obtained using a finite difference solver.
>
> In the table below, we take a $11\times11$ grid, and calculate the relative errors of $K$ and $b$ to provide a quantitative evaluation of the interpretability. Beyond the in-distribution (ID) scenario where the microstructure $b$ is generated using a Gaussian random field with covariance operator $(-\Delta + 5^2)^{-4}$ and the loading field $g$ is generated with $(-\Delta + 5^2)^{-1}$, two out-of-distribution (OOD) scenarios are included: 1) data with ID microstructure $b$ and OOD $g$ using covariance operator $(-\Delta + 5^2)^{-4}$; 2) data with OOD microstructure $b$ using covariance operator $(-\Delta + 5^2)^{-1}$ and ID $g$. Intuitively, these two OOD tasks are more challenging. One can see that NIPS achieves the smallest error in both test errors and kernel/microstructure errors almost in all scenarios.
>
> Table 1: Test errors, kernel errors, and microstructure errors for the Darcy flow problem. Bold numbers highlight the best method.
> |Data Setting | Case | Model | \#param | Test error | Kernel error | Microstructure error |
> | :------------- | :-----------: | :-----------: | :-----------: | :-----------: | :------------- | :-----------: |
> |ID | No noise | NIPS | 327,744 | **4.09\%** | **9.24\%** | 7.92\% |
> | | | NAO | 331,554 | 4.15\% | 10.35\% | **7.09\%** |
> |OOD, Scenario 1 | No noise | NIPS | 327,744 | **3.40\%** | **10.63\%** | **11.23\%** |
> | | | NAO |331,554  | 6.86\% | 14.46\% | 20.37\% |
> |OOD, Scenario 2|No noise | NIPS | 327,744 | **4.98\%** | **9.68\%** | **16.06\%** |
> | | | NAO |331,554  | 5.57\% | 10.94\% | 20.30\% |
> |ID|Noise $\epsilon\sim\mathcal{N}(0,0.01^2)$ | NIPS | 327,744 | 4.40\% | 9.47\% | 7.97\% |
> |OOD, Scenario 1|Noise $\epsilon\sim\mathcal{N}(0,0.01^2)$ | NIPS | 327,744 | 3.88\% | 11.14\% | 13.63\% |
> |OOD, Scenario 2|Noise $\epsilon\sim\mathcal{N}(0,0.01^2)$ | NIPS | 327,744 | 5.65\% | 10.21\% | 17.36\% |
> |ID|Noise $\epsilon\sim\mathcal{N}(0,0.1^2)$ | NIPS | 327,744 | 9.98\% | 21.77\% | 14.84\% |
> |OOD, Scenario 1| Noise $\epsilon\sim\mathcal{N}(0,0.1^2)$ | NIPS | 327,744 | 3.84\% | 21.08\% | 19.87\% |
> |OOD, Scenario 2| Noise $\epsilon\sim\mathcal{N}(0,0.1^2)$ | NIPS | 327,744 | 11.67\% | 22.77\% | 26.54\% |
>
> **Real-world data to demonstrate practical applicability**: We appreciate the reviewer's valuable suggestion. To our best knowledge, most available airfoil flow and climate datasets [1-2] are also generated synthetically by solving PDEs. To address the reviewer's request, we instead consider a real-world data setting, by including additional noise in the dataset. In particular, we perturb the training data with additive Gaussian noise:
> \begin{equation}
>     \widetilde{g}(x) = g(x) + \epsilon(x), \quad \epsilon \sim \mathcal{N}(0, \sigma^2),
> \end{equation}
> where $g(x)$ is the true source field, and $\epsilon(x)$ represents zero-mean Gaussian noise with variance $\sigma^2$. The results are presented using the Darcy flow experiment, with $\sigma=0.01$ and $0.1$. From the results, we can see that NIPS's predictions unavoidably deteriorates under the increased level of observational noises, but they remain robust.
>
> [1] Z Li. Fourier neural operator with learned deformations for PDEs on general geometries. JMLR, 2023
>
> [2] J Gupta. Towards multi-spatiotemporal-scale generalized PDE modeling. TMLR, 2022
>
> **Add model architecture to appendix**: We thank the reviewer's valuable suggestions. Besides Fig 1 and Algorithm 1, we will add an additional section to provide details on the model architecture. We will also release source codes and datasets upon paper acceptance to guarantee clarity and reproducibility of all experiments.
>
> **Interpretability variation across different zero-shot conditions**: As discussed above, we consider three zero-shot generalization scenarios (ID, OOD for loading $g$, and OOD for microstructure $b$), with results listed in the above table. Both NIPS and NAO generalize well across these scenarios in forward operator errors and kernel errors. However, microstructure errors may deteriorate, particularly in scenario 2 (OOD microstructure $b$). This is expected, as the microstructure in this scenario is not only OOD but also rougher, incorporating more fine-grained details. This added complexity makes recovering the ground truth more challenging, especially in an unsupervised learning setting.

---

### Decision · Program_Chairs · 2025-05-01

**Decision:**

Accept (poster)

**Comment:**

NIPS presents a meaningful advance in neural operators, balancing efficiency, interpretability, and generalization. The rebuttal convincingly resolved scalability concerns and contextualized limitations (e.g., symmetry assumptions). Given the strong empirical results, clear technical contribution, and alignment with ICML’s scope, acceptance is warranted. Future work could explore broader PDE classes and theoretical guarantees.